# Quantum Energy Current Induced Coherence in a Spin Chain under Non-Markovian Environments

**DOI:** 10.3390/e24101406

**Published:** 2022-10-01

**Authors:** Arapat Ablimit, Run-Hong He, Yang-Yang Xie, Lian-Ao Wu, Zhao-Ming Wang

**Affiliations:** 1College of Physics and Optoelectronic Engineering, Ocean University of China, Qingdao 266100, China; 2Ikerbasque, Basque Foundation for Science, 48011 Bilbao, Spain; 3Department of Physics, University of the Basque Country UPV/EHU, 48080 Bilbao, Spain

**Keywords:** quantum coherence, energy current, non-Markovian dynamics

## Abstract

We investigate the time-dependent behaviour of the energy current between a quantum spin chain and its surrounding non-Markovian and finite temperature baths, together with its relationship to the coherence dynamics of the system. To be specific, both the system and the baths are assumed to be initially in thermal equilibrium at temperature $T_{s}$ and $T_{b}$, respectively. This model plays a fundamental role in study of quantum system evolution towards thermal equilibrium in an open system. The non-Markovian quantum state diffusion (NMQSD) equation approach is used to calculate the dynamics of the spin chain. The effects of non-Markovianity, temperature difference and system-bath interaction strength on the energy current and the corresponding coherence in cold and warm baths are analyzed, respectively. We show that the strong non-Markovianity, weak system-bath interaction and low temperature difference will help to maintain the system coherence and correspond to a weaker energy current. Interestingly, the warm baths destroy the coherence while the cold baths help to build coherence. Furthermore, the effects of the Dzyaloshinskii–Moriya (DM) interaction and the external magnetic field on the energy current and coherence are analyzed. Both energy current and coherence will change due to the increase of the system energy induced by the DM interaction and magnetic field. Significantly, the minimal coherence corresponds to the critical magnetic field which causes the first order phase transition.

## 1. Introduction

Decoherence and dissipation of a quantum system are a consequence of the interaction between the system and its surrounding environment and have been extensively studied in quantum optics, quantum information, or quantum many-body system. Open systems are difficult to deal with due to the complexity of the reservoirs. Born-Markovian approximation has been used to describe the system dynamics, which assumes that the large reservoir is not altered significantly. In this case, the system loses its information into the bath, and these lost information does not play any further role on the system dynamics. At short and intermediate time scales, considering the memory effects of the environment, it may fails to give a correct description of the dynamics. A non-Markovian quantum master equation is therefore required to faithfully reproduce the system dynamics, especially in this era quantum technology in short-time and/or low temperature has been developed thoroughly [1]. In the non-Markovian case, the lost information can flow back to the system from environment within a certain time [2,3,4,5]. The key feature of environmental non-Markovianity is the distinguishability between any two states, i.e., strong non-Markovianity corresponds to larger information backflow [6]. The lost information can flow back to the system within a certain time The bath-to-system backflow of information will affect the system dynamics and has been investigated from different perspectives such as regeneration of the coherence [7], energy [8,9], and heat [10,11]. And these phenomena have been observed in different experimental setups [12,13,14].

Recently, significant efforts have been devoted to non-Markovian dynamics in various aspects of physics, such as quantum chemistry [15], solid state physics [16], and topological physics [17]. Several methods have also been suggested to formally define and quantify the degree of non-Markovianity of the baths [6,18,19]. Global correlation and local information flows in controllable non-Markovian quantum dynamics is recently studied and the quantum Fisher information and quantum mutual information are demonstrated to be capable of measuring the non-Markovianity for a multi-channel open quantum dynamics [20]. Furthermore, in superohmic environment the non-Markovian recovery of the system dynamics and different initial state trace distance non-monotonicity are found using real-time path integral [21]. Nowadays non-Markovianity has been exploited as resource to improve the quantum state transfer fidelity through a spin chain [22], the adiabatic fidelity [23], or quantum communication protocols [24]. Non-Markovian effects from the point view of information backflow is investigated [11], exchange of information and heat in a spin-boson model with a cold reservoir is examined.

In most of these studies, the system is assumed to be initially in a pure state. However the assumption may not be true because of inevitably inaccurate physical operations, environmental temperature and lingering noises. Furthermore, in a multi-qubit quantum system such as nuclear magnetic resonance, it is difficult to manipulate or detect single qubits and prepare pure states [25]. Thus it is of practical significance and necessary to consider initial mixed states in a quantum process in particular qauntum computation [26,27,28]. In this paper, we consider a general case that the system and the baths are both initially in thermal equilibrium at a certain temperature. We focus on the time evolution of the energy current and coherence of the system in an open system. We use NMQSD approach to investigate the non-Markovian dynamics of the system [29,30,31,32]. It determines the quantum dynamics of open systems by solving the non-Markovian diffusive stochastic Schrödinger equation [33,34]. The effects of the environmental (temperature $T_{b}$, non-Markovianity γ, interaction strength Γ) and system (DM interaction strength $D_{z}$, magnetic field intensity $B_{z}$) parameters are analyzed in warm and cold baths, respectively.

## 2. Formalism

In this section, we review the non-Markovian quantum state diffusion approach (Section 2.1) which will be used in the calculation. We then introduce the spin chain model, the energy current and quantum coherence in Section 2.2, Section 2.3 and Section 2.4.

### 2.1. Non-Markovian Quantum State Diffusion

In open systems, the total Hamiltonian can be written as
(1)$$H_{tot}=H_{s}+H_{b}+H_{int},$$
where $H_{s}$, $H_{b}$ denote the system and bath Hamiltonian, respectively. $H_{int}$ is the interaction Hamiltonian between the system and bath. Suppose the system consists of many qubits. It is reasonable to assume that each qubit is coupled to its own environment. We are thus led to a more complicated model in which the system couples to a collection of independent baths. The Hamiltonian of the bath reads $H_{b}=\sum_{j=1}^{N}H_{b}^{j}$. $H_{b}^{j}=\sum_{k}\omega_{k}^{j}b_{k}^{j†}b_{k}^{j}$ (setting ħ=1) is the Hamiltonian of the *j*th baths with $b_{k}^{j†}$, $b_{k}^{j}$ being the bosonic creation and annihilation operators of the *k*th mode with frequency $\omega_{k}^{j}$. The system-bath interaction Hamiltonian $H_{int}$ is given by
(2)$$H_{int}=\sum_{j,k}\left(f_{k}^{j*}L_{j}^{†}b_{k}^{j}+f_{k}^{j}L_{j}b_{k}^{j†}\right),$$
where $L_{j}$ is the Lindblad operator and it characterizes the couplings between the system and the *j*th bath. $f_{k}^{j}$ is the coupling strength between the system and the *k*th mode of the *j*th bath. Assume that the *j*th bath is initially in a thermal equilibrium state at temperature $T_{j}$
(3)$$\rho_{j}\left(0\right)=e^{-\beta H_{b}^{j}}/Z_{j}.$$ Here $Z_{j}=Tr\left[e^{-\beta H_{b}^{j}}\right]$ is the partition function with $\beta_{j}=1/T_{j}$ (setting $K_{B}=1$).

The open system in the bosonic heat bath satisfies the following NMQSD equation [31,33,35]
(4)$$\frac{\partial }{\partial t}|\psi (t)\rangle =\left[-iH_{s}+\sum_{j}\left(L_{j}z_{j}^{*}\left(t\right)+L_{j}^{†}w_{j}^{*}\left(t\right)-L_{j}^{†}{\bar{O}}_{z^{*}}^{j†}\left(t\right)-L_{j}{\bar{O}}_{w^{*}}^{j}\left(t\right)\right)\right]|\psi (t)\rangle ,$$
where $z^{*}\left(t\right)$, $w^{*}\left(t\right)$ are the stochastic environmental noises, and ${\bar{O}}_{\eta }^{j}\left(t\right)=\int_{0}^{t}\alpha_{\eta }^{j}\left(t,s\right)O_{\eta }^{j}\left(t,s,z_{j}^{*},w_{j}^{*}\right)$. The *O* operator is an operator defined by an *ansatz* $O_{\eta }^{j}\left(t,s,z_{j}^{*},w_{j}^{*}\right)|\psi (t)\rangle =\frac{\delta }{\delta \eta (s)}|\psi (t)\rangle$ (for details, see [33]). It has memory kernel and depends on the nature of noise as well as the form of the coupling between the system and the baths. $\alpha_{\eta }\left(t,s\right)$ is the bath correlation function. The density operator of the system can be recovered from the average of the solutions to the NMQSD equation over all the environmental noises. When the environmental noise strength is weak, the non-Markovian master equation can be written as [36]
(5)$$\frac{\partial }{\partial t}\rho_{s}=-i\left[H_{s},\rho_{s}\right]+\sum_{j}\left\{\left[L_{j},\rho_{s}{\bar{O}}_{z}^{j†}\left(t\right)\right]-\left[L_{j}^{†},{\bar{O}}_{z}^{j}\left(t\right)\rho_{s}\right]+\left[L_{j}^{†},\rho_{s}{\bar{O}}_{w}^{j†}\left(t\right)\right]-\left[L_{j},{\bar{O}}_{w}^{j}\left(t\right)\rho_{s}\right]\right\}.$$

The first term on the right-hand side of Equation (Equation 5) accounts for the coherent unitary evolution, which is ruled by the system Hamiltonian $H_{s}$. The other terms on the right-hand side describe the couplings to the environment. For the bath correlation function $\alpha_{\eta }^{j}\left(t,s\right)$, we choose the ohmic type with a Lorentz-Drude cutoff [37,38,39], whose spectral density is given by $J_{j}\left(\omega_{j}\right)=\frac{\Gamma_{j}}{\pi }\frac{\omega_{j}}{1+{\left(\frac{\omega_{j}}{\gamma_{j}}\right)}^{2}}$. Here $\Gamma_{j}$, $\gamma_{j}$ are dimensionless real parameters. $\Gamma_{j}$ describes the overall environmental noise strength to the system dynamical evolution process, and $1/\gamma_{j}$ represents the memory time of the environment. When $\gamma_{j}$ approaches to zero, the bosonic bath bandwidth is narrow, which corresponds to colored noise, then the environment manifests a strong non-Markovianity. On the contrary, for a large $\gamma_{j}$, the distribution of the Lorentzian spectrum represents a white noise, which corresponds to Markovian limit. ${\bar{O}}_{\eta }^{j}\left(t\right)$ can be numerically calculated by the following equations [40,41]
(6)$$\frac{\partial {\bar{O}}_{z}^{j}}{\partial t}=\left(\frac{\Gamma_{j}T_{j}\gamma_{j}}{2}-\frac{i\Gamma_{j}\gamma_{j}^{2}}{2}\right)L_{j}-\gamma_{j}{\bar{O}}_{z}^{j}+\left[-iH_{s}-\sum_{j}\left(L_{j}^{†}{\bar{O}}_{z}^{j}+L_{j}{\bar{O}}_{w}^{j}\right),{\bar{O}}_{z}^{j}\right],$$
(7)$$\frac{\partial {\bar{O}}_{w}^{j}}{\partial t}=\frac{\Gamma_{j}T_{j}\gamma_{j}}{2}L_{j}^{†}-\gamma_{j}{\bar{O}}_{w}^{j}+\left[-iH_{s}-\sum_{j}\left(L_{j}^{†}{\bar{O}}_{z}^{j}+L_{j}{\bar{O}}_{w}^{j}\right),{\bar{O}}_{w}^{j}\right].$$

### 2.2. Spin Chain

The NMQSD approach provides a general theory to deal with the non-Markovian dynamics of an open quantum system. The system Hamiltonian can be taken as different forms for different physical systems. The spin chain model has attracted much attention in experimental and theoretical studies due to its rich and exquisite mathematical structure. It is not just an abstract theoretical model but in fact accurately describe the dominant physical phenomena of metals and crystals like ferromagnetism and antiferromagnetism [42,43,44,45]. Here in this paper, we take a one-dimensional XY spin chain with DM interaction and external magnetic field. For the individual bath model, each spin is immersed in its own baths (see Figure 1). The Hamiltonian reads
(8)$$H_{s}=\sum_{j=1}^{N}\left[J\left(\sigma_{j}^{x}\sigma_{j+1}^{x}+\sigma_{j}^{y}\sigma_{j+1}^{y}\right)+D_{z}\left(\sigma_{j}^{x}\sigma_{j+1}^{y}-\sigma_{j}^{y}\sigma_{j+1}^{x}\right)+B_{z}\sigma_{j}^{z}\right],$$
where $\sigma_{j}^{\alpha }$(α=x,y,z) represents the α component of the Pauli matrix for spins and *J* is the coupling constant between the nearest-neighbour sites. *N* is the number of site and we assume the periodic boundary conditions $\sigma_{N+1}^{\alpha }=\sigma_{1}^{\alpha }$. The parameters $D_{z}$ and $B_{z}$ are DM interaction and uniform magnetic field strength. Note here we consider z-component DM interaction $D_{z}$ and uniform magnetic field $B_{z}$ along *z* direction. Antiferromagnetic spin chain have gained increasing attention in spin technology owing to their advantages over their ferromagnetic counterpart in considerable spin orbit, achieving ultrafast dynamics, and large magnetoresistance transport [46,47,48]. For this model, we take antiferromagnetic coupling J=1 throughout and $0\leq D_{z}\leq 1$.

Now we assume that initially the spin chain is also at thermal equilibrium, with the density matrix $\rho_{s}\left(0\right)=e^{-\beta_{s}H_{s}}/tr\left(e^{-\beta_{s}H_{s}}\right)$. $\beta_{s}=1/T_{s}$ is the inverse temperature. The high-temperature approximation can be taken when $∥H_{s}∥\ll T_{s}$$\left(∥H_{s}∥=tr\sqrt{H_{s}^{†}H_{s}}\right)$. In this case, $\rho_{s}\left(0\right)$ can be aprroximately expressed by the first two terms of the Taylor expansion [25]
(9)$$\rho_{s}\left(0\right)=\frac{1}{2^{N}}\left(I-\frac{H_{s}}{T_{s}}\right),$$
where *I* is the identity matrix of dimension $2^{N}$. Although the thermal equilibrium state is highly mixed, experimental and theoretical studies have shown that this state can be transformed into a pseudo-pure state [49,50]
(10)$$\rho_{s}\left(0\right)=\frac{1}{2^{N}}\left(1-\epsilon \right)I+\epsilon |\varphi_{0}\rangle \langle \varphi_{0}|.$$

Pseudo-pure state is still a mixed state ($tr(\rho_{s}^{2})<1$), but in the whole evolution the state $|\varphi_{0}\rangle$ appears with probability $\left(1-\epsilon \right)/2^{N}+\epsilon$ and it can carry out some manipulations and quantum algorithms designed for pure states [51]. All of the states orthogonal to state $|\varphi_{0}\rangle$ appear with equal probabilities of $\left(1-\epsilon \right)/2^{N}$, where the coefficient ϵ is usually small. This pseudo-pure state technique provides a convenient starting point for quantum information processing with less than 10 qubits [52].

For the initial density operator of the system, according Equation (Equation 10) throughout the paper we take N=4, and assume
(11)$$\begin{array}{c} |\varphi_{0}\rangle =\left(|1000\rangle +|0100\rangle +|0010\rangle +|0001\rangle \right),\;\;\;\;\;\;\;\;\;\;\;\;\;\;\;\;\;\;\;\;\;\;\;\;\epsilon =-3\beta_{s}. \end{array}$$

Note that the temperature-dependent parameter ϵ→0 in the high-temperature limit and the initial density matrix is more inclined to be a mixed state $\rho_{s}\left(0\right)\to \frac{1}{2^{N}}I$.

### 2.3. Energy Current

The energy transfer between the system and the environment is important in the study of thermodyanmic properties of an open system. The exchange energy between the open system and environment is accompanied by the exchange of entropy, which is one of the important criteria to evaluate the amount of information stored in a quantum system. Therefore, energy current can indirectly describe the information storage capacity of the environment. Recently, an exactly solvable model was proposed to investigate the quantum energy current between a nonlinearly coupled bosonic bath and a fermionic chain [53]. The adiabatic speedup and the associated heat current with and without pulse control is investigated, where the heat current is defined as the difference of the energy current and the power [10,54]. The energy current can be defined as the derivative of the expectation value of $H_{s}$ [55,56]
(12)$$E\left(t\right)=\frac{\partial }{\partial t}tr\left[\rho_{s}H_{s}\right].$$

The above definitions has been proved to be valid for a non-equilibrium spin—boson model and a three-level heat engine model in the case of non-perturbative and non-Markovian conditions [57], where the reduced hierarchal equations of motion approach is used.

### 2.4. Quantum Coherence

Quantum coherence or quantum superposition lies at the hotspot of quantum theory, and it is a very valuable resource for quantum information processing [58,59]. It is also of equal importance as entanglement in the studies of both bipartite and multipartite quantum systems [60]. Based on the framework of consistent resource theory, the commonly used coherence measure is the $l_{1}$ norm coherence, which is a sum of all off-diagonal elements of the density matrix [61]
(13)$$C\left(\rho \right)=\sum_{a\neq b}\left|\rho_{a,b}\right|.$$

## 3. Numerical Results and Discussions

Based on the definition of energy current and quantum coherence in Equations (Equation 12) and (Equation 13), we next numerically calculate the non-Markovian dynamics of the energy current and quantum coherence. Now the model is that each spin is immersed in its individual bath [22]. However, due to the neighbor spins are close to each other, we assume the same environmental parameters $\Gamma =\Gamma_{j}$, $\gamma =\gamma_{j}$, $T_{b}=T_{b}^{j}$ for all these *j*th baths. We also assume there is no initial system-bath correlations, $\rho \left(0\right)=\rho_{s}\left(0\right)⨂\rho_{b}\left(0\right)$. $\rho_{s}\left(0\right)$ is often taken as pure state, and $\rho_{b}\left(0\right)$ is in a vacuum state [22], or thermal equilibrium state [62,63]. As an example, throughout the paper we consider the quantum dissipation model, in this case the Lindblad operator $L_{j}=\sigma_{j}^{-}$. $\sigma_{j}^{-}=\left(\sigma_{j}^{x}-i\sigma_{j}^{y}\right)/2$. In this case, the number of excitations is not conserved, and transitions between different subspaces with certain number of excitations occur [64]. We will study the behavior in time of the energy exchange between the system and the baths and the quantum coherence of the system under the influence of the baths.

We first explore the effects of non-Markovianity, environmental temperature and noise strength on the system dynamics when the system couples to warm baths ($T_{b}>T_{s}$). In Figure 2, we plot the energy current as a function of time *t* for different parameter γ (Figure 2a), $T_{b}$ (Figure 2b) and Γ (Figure 2c), respectively. In the inset of Figure 2 we also plot the corresponding coherence dynamics. In Figure 2, we take $T_{s}=10$ and the weak coupling limit Γ=0.003, $T_{b}=80$ for Figure 2a, γ=5,Γ=0.003 for Figure 2b, $T_{b}=80,\gamma =5$ for Figure 2c. From Figure 2, we can see that the energy current between the system and baths increases with increasing parameters γ, Γ and $\left|T_{s}-T_{b}\right|$. That is to say, more Markovian baths, stronger system-baths interactions and higher temperature difference correspond to bigger energy current, which is in accordance with the case that the initial states of the system is in a pure state [54]. Correspondingly, coherence decreases with increasing parameters γ, Γ and $\left|T_{s}-T_{b}\right|$. As expected, non-Markovian baths, weak system-bath interactions and low temperature difference will be helpful to maintain the coherence of the system. Note that in most cases the energy current is positive, which indicates the energy transfer from environment to the system. At time t=0, the energy current is 0. In a short time region, the energy starts to increase and reach a peak value. Then it decreases in long time region. For a relatively strong non-Markovian bath (Figure 2a γ=0.5), the energy current exhibits a oscillation pattern before it reaches steady state, which has negative values (from system to bath). In this case, the coherence also shows an osillation, i.e., the energy backflow from sytem to baths affects the coherence of the system.

Next we discuss a contrary case that the system is immersed in cold baths ($T_{s}>T_{b}$). Figure 3 again plots the effects of the parameters γ, $T_{b}$ and Γ on the energy current and coherence. Here we take a high system temperature $T_{s}=100$, clearly the coefficient ϵ in Equation (Equation 11) becomes smaller, pseudo-pure state purity decreases, thus weakening the quantum coherence in the initial state (the initial coherence is now 0.09 from Figure 3). Compared with Figure 2, we find that the same conclusion is obtained that the energy current increases with increasing parameters γ, Γ and temperature difference $\left|T_{s}-T_{b}\right|$. But a negative energy corresponds to the energy transfer from a warm system to the cold baths. During the calculation, we find that initially positive energy current occurs in a very short time, these initial currents reflect the response of the system to instantaneous coupling to the baths at time t=0. For the coherence, the conclution in Figure 2 also holds: non-Markovian baths, weak system-bath interactions and low temperature difference will be helpful to maintain the coherence of the system. But surprisingly, the cohercence increases with increasing parameter γ and Γ but decreases with increasing $T_{b}$. That is to say, for warm system in cold baths, more Markovian, lower temperature, and stronger interaction strengths helps the system to be a more pure state. This phenomenon can be explained as follows: when a small warm system is surrounded by large cold baths, the system energy dissipates into the bath quickly and the system gets cool down, thus the coherence starts to increase due to low system temperature.

The DM interaction is an antisymmetric exchange interaction between nearest site spins, arising from spin-orbit coupling. It emerges in Heisenberg model lacking inversion symmetry and promotes noncollinear alignment of magnetic moments and induces chiral magnetic order [65,66]. Although this interaction is weak, it has many spectacular features, for example, chiral Néel domain walls [67,68], skyrmions [69], etc, implying that a study of spin models with DM interaction could have realistic applications. In antiferromagnetic materials, DM interaction will break the antiparallelism of the spin chain spatial structure. This change enriches the physical properties of antiferromagnetic materials [70,71], such as in coupled quantum dots in GaAs [72]. Next we will discuss the effects of DM interaction on the energy current and coherence. Figure 4 plots the quantum coherence and energy current dynamics for different DM interaction strength $D_{z}$ in the warm baths ($T_{b}=80$, $T_{s}=20$) and cold baths ($T_{b}=20$, $T_{s}=80$), respectively. Other parameters are taken as Γ=0.005, γ=2, $J=1,B_{z}=J$. From Figure 4a, for warm baths the negative energy current is obtained by the introduction of DM interaction. Strong DM interaction strength $D_{z}$ restrains the positive energy current and enlarges the negative energy current. This is due to, as the DM interaction strength increases, strong spin-orbit couplings cause the neighboring spins inverse antiparallel structures to intersect and the system energy is enhanced, as a result it restrains the energy current from the bath to system and enlarges the reversed current. For the cold baths plotted in Figure 4b, the negative energy current always exists and clearly the energy current increases with increasing $D_{z}$, which is also caused by the increasement of the system energy. From the inset of Figure 4a,b, the coherence of the system decreases with increasing $D_{z}$. Stronger DM interaction will destroy more coherence of the system, i.e., the system energy increase is not conductive to the preservation of quantum coherence, whether in warm or cold baths. In addition, we find that after the evolution time t>4, the quantum coherence in warm bath and cold bath has a significant recovery, which is caused by the non-markovianity of the environment.

At last, we consider the effects of the external magnetic field, which can also affect the spatial structure of spin chain and show a positive aspect in the study of quantum entanglement and quantum state transport in a spin chain [73,74,75]. In Figure 5, we plot the energy current and coherence dynamics for different external magnetic field intensity $B_{z}$ in warm baths and cold baths, respectively. The parameters are the same as in Figure 4 except that $D_{z}=0.3$. First from Figure 5a for the warm bath case, the positive energy current decreases with increasing $B_{z}$ for a weak magnetic field ($B_{z}=J$). When $B_{z}=2J$, the energy current starts to reverse and it increases with increasing $B_{z}$. The coherence in the inset of Figure 5a also shows this decrease-increase behavior. $B_{z}=2J$ corresponds to the lowest coherence. Why strong field can cause the reverse of the energy current? From Figure 5a the energy transfer from the low temperature system to the high temperature baths always occurs in a strong external field (e.g., $B_{z}=5J$). The spin chain is more inclined to be at antiferromagnetic order in thermal equilibrium, but the introduction of magnetic field reduce the antiferromagnetic order. When the external magnetic field increases to the critical field point ($B_{z}=2J$), the spin chain polarization flips into the direction perpendicular to the field, and the phase transition characteristics are immediately captured by the evolutionary properties of coherence or the energy current. The spin-flip transition of antiferromagnetic materials under the external magnetic is a first-order quantum phase transition, and can be observed experimentally [76,77]. Strong field causes the spin parallel to the direction of the field and corresponds to a high potential energy, thus the energy current from the low temperature system to high temperature baths occurs. Strong field also corresponds to high coherence and weakens the decoherence of the system. The increase of the energy caused by the field can also fairly explain the results in Figure 5b. The negative energy current always increases with increasing $B_{z}$ for cold baths. The energy difference between system and baths enlarges the energy current. In this case, the phase transition ($B_{z}=2J$) can not be characterized by the energy current reverse, but it can still be characterized by the coherence.

## 4. Conclusions

We have investigated the energy current and coherence dynamics in open systems. The system is a one dimensional spin chain with periodic boundary conditions. We have considered the independent bath model, i.e., each spin is immersed in its own non-Markovian bath. Specifically, the spin chain is initially at thermal equilibrium at finite temperature, or equivalently at pseudo-pure state. By using the NMQSD approach, we calculate the energy current between the system and baths and the coherence dynamics in warm baths and in cold baths, respectively. The effects of the bath non-Markovianity, bath temperature and system-bath coupling strength on the energy current and coherence are analyzed. We find that non-Markovianity, low temperature difference and weak coupling correspond to weaker energy current and are in favour of the coherence for both warm and cold baths. However, the coherence will be damaged by the warm baths but in cold baths it can be significantly enlarged. Cold environment will help to boost the coherence. We also consider the influences of the DM interaction on the energy current and coherence. The DM interaction will increase the system energy for antiferromagnetic chain. Then it shows different behaviours for warm and cold baths. For warm baths, strong DM interactions restrain the positive energy current and enlarge negative energy current. For cold baths, it only exists negative energy current, and strong DM interactions also enlarge the negative energy current. The coherence will always decreases with the DM interaction strength $D_{z}$. Finally we have also studied the magnetic field effects, where $B_{z}=2J$ is a critical value which corresponds to the first a quantum phase transition. The magnetic field can also increase the system energy, then similar as the DM interaction case, for warm baths, strong magnetic fields restrain the positive energy current and enlarge negative energy. For cold baths, strong magnetic fields also enlarge the negative energy currents. It is interesting to note that for both types of baths the coherence demonstrates decrease-increase behaviour with increasing $B_{z}$, and the lowest coherence corresponds to the critical value $B_{z}=2J$. These investigations, based on microscopic understanding, elucidates the relation of energy current and quantum coherence, which might potentially be a good reference in context of quantum thermodynamics of non-Markovian open quantum systems [78], as well as in study of environment-induced quantum coherence [79,80,81].

## Figures and Tables

**Figure 1 entropy-24-01406-f001:**
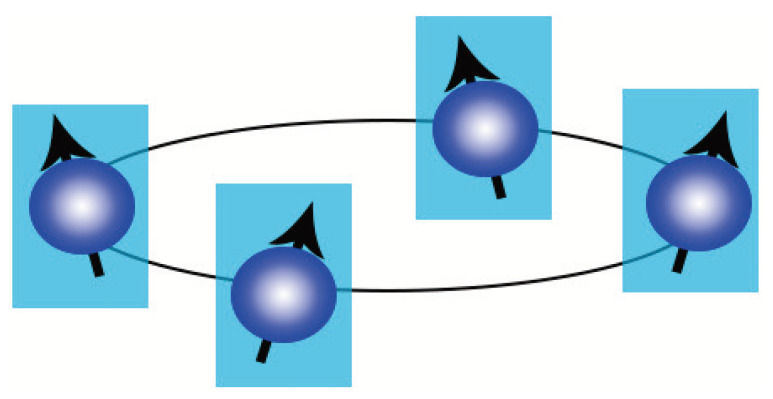
(Color on line) The sketch of the spin chain. Each spin is immersed in its own non-Markovian and finite temperature heat bath.

**Figure 2 entropy-24-01406-f002:**
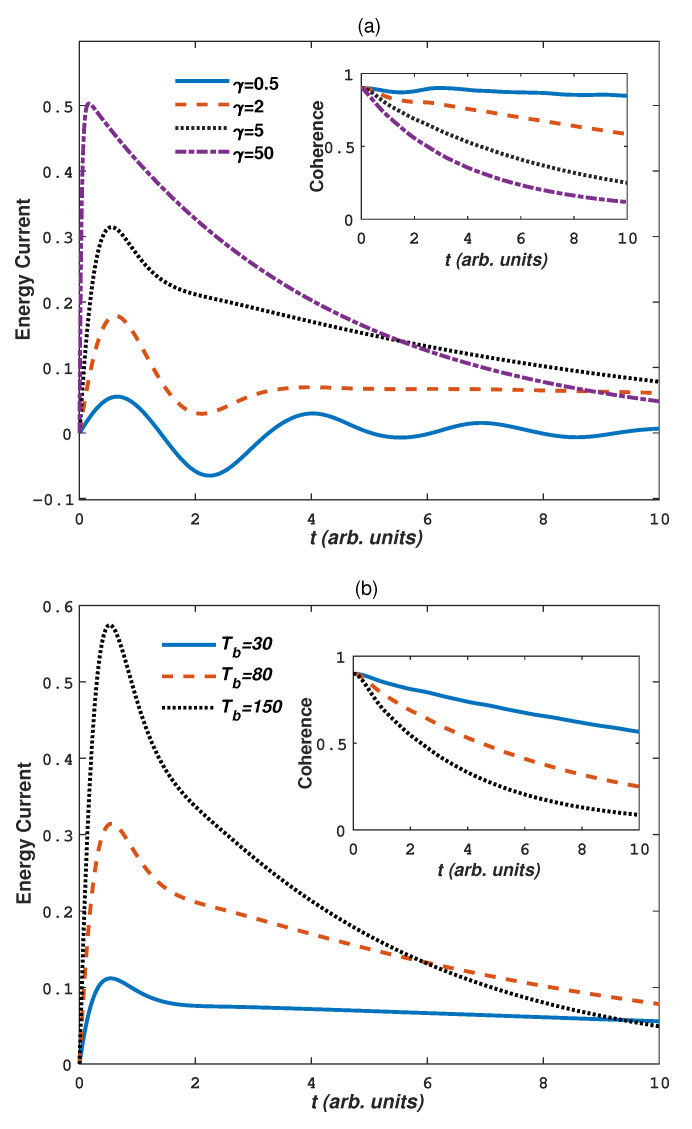
(Color on line) The energy current and quantum coherence as a function of time *t* in warm baths ($T_{s}<T_{b}$) for different values of bath parameters: (**a**) γ, $T_{b}=80$, Γ=0.003; (**b**) $T_{b}$, γ=5, Γ=0.003; (**c**) Γ, $T_{b}=80$, γ=5. Other parameters are take as $T_{s}=10$, J=1, $D_{z}=0$ and $B_{z}=0$.

**Figure 3 entropy-24-01406-f003:**
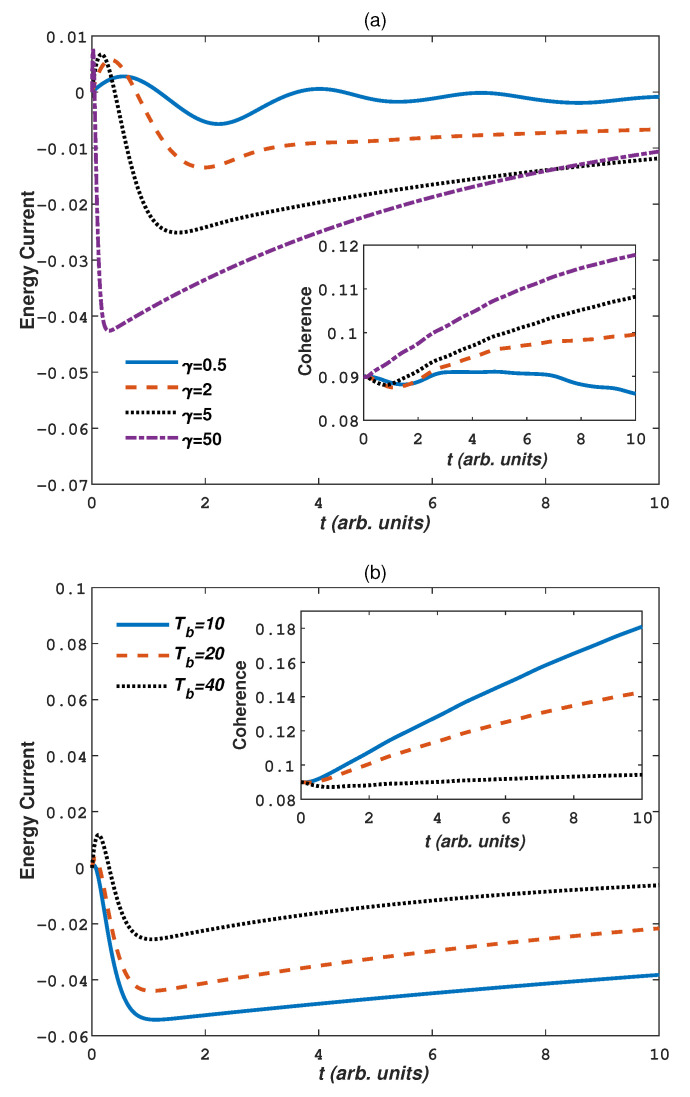
(Color on line) The energy current and quantum coherence as a function of time *t* in cold baths ($T_{s}>T_{b}$): (**a**) γ, Γ=0.005, $T_{b}=10$; (**b**) $T_{b}$, γ=10, Γ=0.005; (**c**) Γ, γ=10, $T_{b}=10$. Other parameters are $T_{s}=100$, J=1, $D_{z}=0$, and $B_{z}=0$.

**Figure 4 entropy-24-01406-f004:**
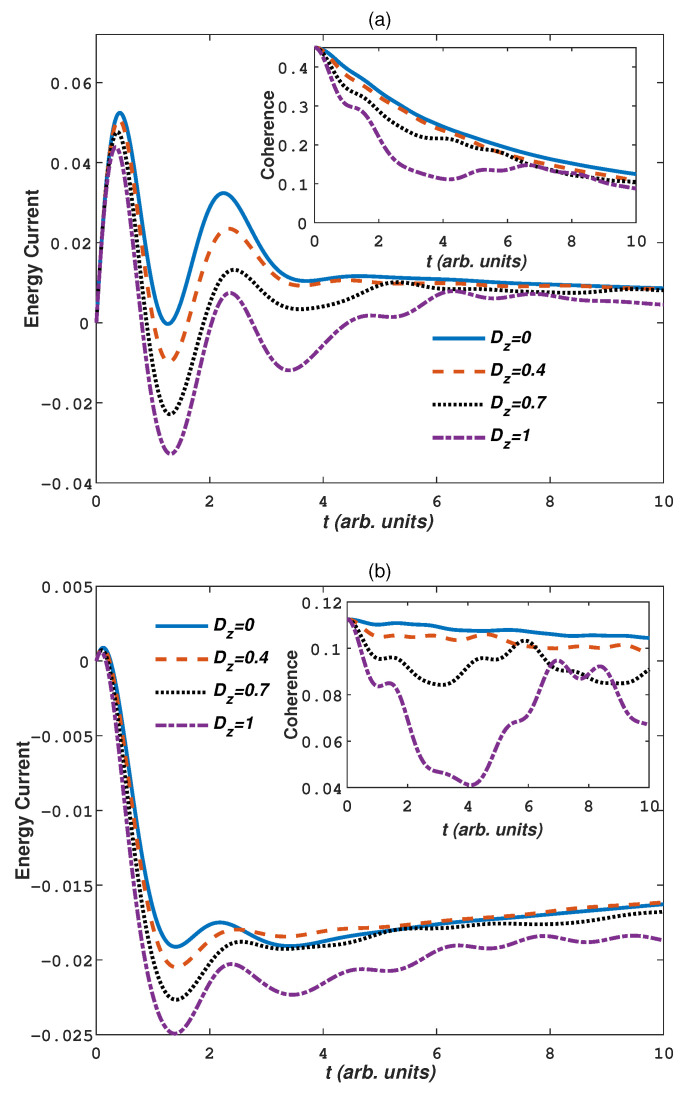
(Color on line) The dynamics of the energy current and quantum coherence with different DM interaction strength $D_{z}$ in (**a**) warm baths ($T_{s}=20,T_{b}=80$) and (**b**) cold baths ($T_{s}=80,T_{b}=20$). Other parameters are $B_{z}=J$, γ=2, Γ=0.005, J=1.

**Figure 5 entropy-24-01406-f005:**
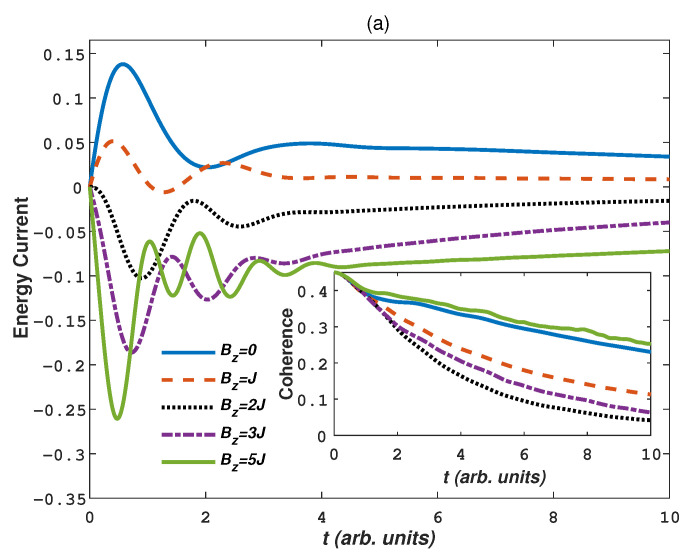
(Color on line) The dynamics of energy current and quantum coherence for different $B_{z}$ in (**a**) warm baths ($T_{s}=20,T_{b}=80$); (**b**) cold baths ($T_{s}=80,T_{b}=20$). Other parameters are N=4, γ=2, Γ=0.005, J=1, $D_{z}=0.3$.

## Data Availability

The datasets used and/or analyzed during the current study are available from the corresponding author on reasonable request.

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
