# Peer review of "Quantum Energy Current Induced Coherence in a Spin Chain under Non-Markovian Environments"

_entropy, 2022, doi:10.3390/e24101406_

Round 1

Reviewer 1 Report

Please see the attached report for more details. The report can be forwarded to the author.

Author Response

Response to Reviewer 1:

Dear Referees:

Many thanks to the Referees for encouraging our work. We also thank you very much for your useful comments and professional advice for clarifying, improving and correcting some parts in the paper. These opinions help to improve academic rigor of our article. Based on your suggestion and request, we have carefully revised the paper and made corrected modification on the revised manuscript, as explained below. We hope that our modifications meet the requirements of the Referees.

The corresponding changes have been marked red in the pdf.

Specific Reply to Technical comments

 Mathematical formalism:

Point: Sec. 2.3. provides a very brief definition of the energy current. However, this quantity appears crucial for the study. I advise the authors to elaborate more on this key figure. For example, is the energy backflow somehow connected to the information backflow that is a common indicator of non-Markovianity (according to the BLP measure)?

 Response : Thanks for the suggestion.  According to the suggestions, we have carefully reviewed related literature and conducted some research. We find it is hard to give a specific correlation between energy current and information current. We will definitely consider this question in our future research. We have added (line 144 to 148)  “The exchange energy between the open system and environment is accompanied by the exchange of entropy, which is one of the important criteria to evaluate the amount of information stored in a quantum system. Therefore, energy current can indirectly describe the information storage capacity of the environment.” in our resubmited manuscript.

We also added "The above definitions has been proved to be valid for a non-equilibrium spin–boson model and a three-level heat engine model in the case of non-perturbative and non-Markovian conditions \cite{A. Kato, Y. Tanimura, J. Chem. Phys. 145 (2016) 224105.}, where the reduced hierarchal equations of motion approach is used." below Eq.(12).“ in line 154 to157.

Presentation of the results:

Point 1: Most of all, please notice that the figures are too stretched horizontally, which makes it difficult to properly observe and analyze the tendencies of selected figures of merit. Please provide figures in the optimal resolution.

Response 1: Thanks, we have provided the best resolution figures in the revised manuscript.

Point 2: The energy current is investigated in the time domain, but the plots do not give any specific unit for time. If time is considered in an arbitrary unit, it should be noted in the axis label (arb. units).

Response 2: Thanks. The time t in our paper is units, which we have marked in the axis label.

References:

Point 1: It is not possible to follow the references because instead of corresponding numbers, we can only see question marks in the text. Consequently, I am not able to fully evaluate the analysis and discussion since I cannot track which particular publication is being referred to. Please amend the citations.

Response 1: We apologize to the reviewer for the incorrect use of the template that caused the references not to follow. We corrected this problem in the resubmitted manuscript.

Point 2: By looking at the list of references, I see that the authors have included many celebrated works on open quantum systems, particularly non-Markovian dynamics. However, I think the authors failed to pick up recent papers on dynamics of open quantum systems, including the articles published in MDPI journals.

Response 2: Thanks to our careful reviewer. As instructed by the reviewer, we reviewed recent papers on open quantum systems, especially high-quality papers published by MDPI, such as Entropy, Symmetry, and Mathematics. Among them, we found a lot of high-quality papers about open quantum systems, spin chains, quantum communication and quantum coherence. A lot of inspiration and theoretical knowledge have been obtained, and cited in this paper.

Conclusions:

Point: I believe the manuscript has the potential for publication in Entropy. If the authors respond to the issues indicated in the report, I will eagerly read the manuscript again to evaluate the improvement.

Response: We thank our referees again for his/her helpful comments and suggestions. These suggestions will help us improve the quality of our paper. We have carefully revised our manuscript according to the comments. And we sincerely hope that the reviewer will review our work again.

Reviewer 2 Report

In this manuscript, Ablimit et al. simulate the open quantum system dynamics of a periodic spin chain using the non-Markovian quantum state diffusion approach. Analysis of the energy current and l1 norm coherence measure on varying the system-bath coupling strength, system-bath temperature difference and bath relaxation rate (proportional to the lifetime of non-Markovian memory effects) shows that system coherence is sustained by more weakly coupled, non-Markovian baths at temperatures close to that of the system. The authors then further consider the effect of the DM exchange interaction, which decreases system coherence, and an external magnetic field, for which a minimum in the system coherence identifies a phase transition. In this way the authors provide valuable insight and reference simulations for the design of systems which maintain prolonged coherence, essential for the development of modern quantum information technologies. I find their direct comparison of warm vs. cold baths with respect to the system temperature particularly interesting and informative. I therefore consider this work a valuable contribution to this special issue.

My major comments are as follows:

The caption of figure 3 suggests the solid blue line in (b) and the dashed orange line in (c) have the same parameters and yet the magnitude of the results are very different. Perhaps these are mislabelled?

As the authors discuss, it is clear from figure 4 that coherence decreases with increasing Dz, but it is interesting in figure 4(b) that the purple line for Dz=1 shows a significant revival in coherence for t>4 which perhaps deserves greater comment. Have the authors investigated this any further, possibly in connection with the greater non-Markovianity in these simulations?

In several places (e.g. lines 227, 250, 270, 277) the authors discuss the "improvement" of the system energy, which is ambiguous in my opinion and it would be better to more accurately state the "increase" in system energy.

I also find the "decrease-increase-decrease" behaviour discussed on lines 237 and 280 unclear. I agree that figure 5(a) shows a decrease in the coherence to a minimum when Bz=2J before increasing again, but see no second decrease. The authors ought to rephrase this to clarify their interpretation.

Other specific comments:

In equation (8) there may be a misplaced bracket as currently the final term for the external field at site j includes the nearest-neighbour coupling constant J, which is not included in the references Liu et al. Phys. Rev. A, 83, 052112 (2011) and Abliz et al. Phys. Rev. A, 74, 052105 (2006) given by the authors, for example, and seems odd given that later Bz = J. 

Line 91: "spectrum density" is more commonly known as the "spectral density".

Line 177-178: Instead of "low temperature bath" I believe they more specifically mean "low temperature difference". 

Line 272: "enlarge negative energy current" rather than "enlarge negative energy".

Author Response

Response to Reviewer 2:

General Reply

Dear Referees:

Many thanks to the Referees for encouraging our work. We also thank you very much for your useful comments and professional advice for clarifying, improving and correcting some parts in the paper. These opinions help to improve academic rigor of our article. Based on your suggestion and request, we have carefully revised the paper and made corrected modification on the revised manuscript, as explained below. We hope that our modifications meet the requirements of the Referees.

The corresponding changes have been marked red in the pdf.

Specific Reply to Technical comments

 Major Comments:

Point 1: The caption of figure 3 suggests the solid blue line in (b) and the dashed orange line in (c) have the same parameters and yet the magnitude of the results are very different. Perhaps these are mislabelled?

 Response 1: Thanks for our careful referee.  It is an error. We apologize and have corrected this issue in the resubmitted manuscript.

 Point 2: As the authors discuss, it is clear from figure 4 that coherence decreases with increasing Dz, but it is interesting in figure 4(b) that the purple line for Dz =1 shows a significant revival in coherence for t>4. Which perhaps deserves greater comment. Have the authors investigated this any further, possibly in connection with the greater non-Markovianity in these simulations?

Response 2: Good suggestion. We have studied the phenomenon of increased coherence for t>4. As the reviewer said, this is caused by the non-Markovian nature of the environment. We add an explanation for this phenomenon in our paper (line 241 to 243).

 The details are as follows:

 “In addition, we find that after the evolution time t > 4, the quantum coherence in warm bath and cold bath has a significant recovery, which is caused by the non-markovianity of the environment.”

Point 3: In several places (e.g. lines 227, 250, 270, 277) the authors discuss the “improvement” of the system energy, which is ambiguous in my opinion and it would be better to more accurately state the “increase” in system energy.

 Response 3: Thanks. We have changed“improvement” to“increase”in lines 14, 227 (now line 240), 250 (now line 265), 270 (now line 285), 277 (now line 292) according to the comments.

Point 4: I also find the “decrease-increase-decrease” behaviour discussed on lines 237 and 280 unclear. I agree that figure 5(a) shows a decrease in the coherence to a minimum when Bz=2J before increasing again, but see no second decrease. The authors ought to rephrase this to clarify their interpretation.

Response 4: Thanks. This is indeed an error caused by our carelessness. Now we have changed “decrease-increase-decrease behaviour”to “decrease-increase behaviour”in line 237 (now line 252) and 280 (now line 295), according to your suggestion.

Other Specific Comments:

Point 1: In equation (8) there may be a misplaced bracket as currently the final term for the external field at site j includes the nearest-neighbour coupling constant J, which is not included in the references Liu et al. Phys. Rev. A, 83, 052112 (2011) and Abliz et al. Phys. Rev. A, 74, 052105 (2006) given by the authors, for example, and seems odd given that later Bz= J

Response 1: Thanks. This is indeed an error. Now we have modified it according to your suggestion. 

Point 2: Line 91: “spectrum density”is more commonly known as the“spectral density”.

Response: Thank s. We have changed“spectrum density” to“spectral density”in line 91(now line 96) according to the comments.

Point 3: Line 177-178: Instead of “low temperature bath” I believe they more specifically mean “low temperature difference”.

Response: Thanks. We have changed“low temperature bath” to“low temperature difference”in line 177-178 (now line 191) according to the comments.

Point 4: Line 272: “enlarge negative energy current”rather than“enlarge negative energy”.

Response: Thanks. We have changed “enlarge negative energy”to“enlarge negative energy current” in line 272 (now line 287) according to the comments.

Reviewer 3 Report

The current manuscript presented by Ablimit et. al reports the quantum energy transport and the coherence in a spin chain in presence of a non-markovian bath. The time-resolved dynamics are performed with non-Markovian quantum state diffusion (NMQSD) equation approach. The authors analyzed their results in different bath parameters, including the system-bath interactions, the temperature of the bath, etc. Their analysis shows that the warm bath destroys the coherence whereas the cold bath helps to build the coherence. The author further analyzed the effects of the Dzyaloshinskii–Moriya (DM) interaction and the effects of coherence in the applied external magnetic field. The manuscript is well-written and explicit, and the conclusions have supported the calculations. I thus accept this manuscript after making the following changes.

1. The author should mention what is the meaning of non-Markovianity in the first place as it might not be clear to the broader audience. 

2. In the introduction the author mentioned some quantum dynamics methodologies that can incorporate the non-Markovianity of bath effects. I think some recent formulation of real-time path-integral studies has this potential. The author should mention them as well.

3. The bath considered in this work is a bosonic bath. I am wondering what will be the outcome if one considers the fermionic bath. Can the author comment on that?

4. What will be the validity of the argument "Now the model is that each spin is immersed in its individual bath. However, due to the neighbor spins being close to each other, we assume the same environmental parameters". Can the author elaborate on what they mean by close to each other?

5. For fig.2 a,b, and c can I think about for \gamma= 50, T_b = 150, and \Gamma = 0.009 the dynamics is purely Markovian?

6. I am not fully convinced by the statement that "the stronger DM interaction destroys more coherence", in other words, the dynamics will be less coherent. Is that so? The reason I am confused is that if I see fig. 4a at t = 6-10 the purple line (D_z = 1.0) is still showing oscillation whereas the blue line (D_z = 0) is already reached the steady state current. Similar results show in fig.4b as well. Can the author comment on this?

7. The future perspective and the immediate significance of this study needs to be elaborated. 

Author Response

Response to Reviewer 3:

General Reply

Dear Referees:

Many thanks to the Referees for encouraging our work. We also thank you very much for your useful comments and professional advice for clarifying, improving and correcting some parts in the paper. These opinions help to improve academic rigor of our article. Based on your suggestion and request, we have carefully revised the paper and made corrected modification on the revised manuscript, as explained below. We hope that our modifications meet the requirements of the Referees.

The corresponding changes have been marked red in the pdf.

Specific Reply to Technical comments

Point 1: The author should mention what is the meaning of non-Markovianity in the first place as it might not be clear to the broader audience. 

Response 1: Thank you for your suggestion. We have added to the manuscript (line 30 to 33) a brief introduce of non-Markovianity.

 The details are as follows:

“In the non-Markovian case, the lost information can flow back to the system from environment within a certain time. The key feature of environmental non-Markovianity is the distinguishability between any two states, i.e., strong non-Markovianity corresponds to larger information backflow.”

Point 2:  In the introduction the author mentioned some quantum dynamics methodologies that can incorporate the non-Markovianity of bath effects. I think some recent formulation of real-time path-integral studies has this potential. The author should mention them as well.

Response 2: Thanks you for your constructive suggestions. We have mention the real-time path-integral studies in the introduction (line 45 to 47) .

The details are as follows:

“Furthermore, in superohmic environment the non-Markovian recovery of the system dynamics and different initial state trace distance non-monotonicity are found using real-time path integral.”

Point 3: The bath considered in this work is a bosonic bath. I am wondering what will be the outcome if one considers the fermionic bath. Can the author comment on that?

Response 3: The question raised by the referee is really a good one on which we have previously not very focused closely. Due to the anti-easy nature of the fermionic bath, the QSD equations in the fermionic bath are more complicated than in the bosonic bath, especially in the multi-bath, multi-qubit model that we study, it is difficult to give the exact QSD equations, so at present we cannot precisely state the dynamical properties of the model in the fermionic bath. However, based on the study of bosonic,fermionic baths(Phys. Rev. A. 84, 032101 (2011),Phys. Rev. A. 86, 032116 (2012)) and the results in some simple models we calculated, we can determine that the effect of environmental non-Markovianity, bath temperature and coupling strength is similar to that of the bosonic bath. But, the effect of the external magnetic field and DM interaction are to be studied, and this will be one of our future research directions.

Point 4: What will be the validity of the argument "Now the model is that each spin is immersed in its individual bath. However, due to the neighbor spins being close to each other, we assume the same environmental parameters". Can the author elaborate on what they mean by close to each other?

Response 4: Due to the small size of quantum devices, the distance between adjacent sites on the spin chain is also nano-scale, so their surrounding environment is similar to share the same environmental parameters.

Point 5: For fig.2 a,b, and c can I think about for γ= 50, Tb = 150, and Γ = 0.009 the dynamics is purely Markovian?

Response 5: In our model, the environmental non-Markovianity only depends on γ. Therefore, γ=50 can be viewed as a pure Markovian dynamics. As noted by careful reviewers, the system dynamics also exhibits Markovianity in the case of Γ=0.09 3 and Tb=150. This is due to, the quantum fluctuations caused by the strong coupling strength and high temperature enhance the information dissipation in the reflux process, thus showing the Markovian dynamics. It is not the shortening of environmental memory time that leads to the Markovian case.

Point 6. I am not fully convinced by the statement that "the stronger DM interaction destroys more coherence", in other words, the dynamics will be less coherent. Is that so? The reason I am confused is that if I see fig. 4a at t = 6-10 the purple line (Dz = 1.0) is still showing oscillation whereas the blue line (Dz = 0) is already reached the steady state current. Similar results show in fig.4b as well. Can the author comment on this?

Response 6: Thank you for your attention on this. Although, when Dz=1, there will be oscillations in t=6-10, while Dz=0 has reached the steady state. However, we can see that, the coherence is lower when DZ=1 than DZ=0 during the whole evolution time. Through further study, we found that this oscillation is caused by the non-Markovianity of the environment. In the revised manuscript, we add an explanation of this phenomenon in lines 241 to 243.

 The details are as follows:

 “In addition, we find that after the evolution time t>4, the quantum coherence in warm bath and cold bath has a significant recovery, which is caused by the non-markovianity of the environment.”

Point 7: The future perspective and the immediate significance of this study needs to be elaborated. 

Response 7: Thank you for your suggestion. Besides the elaboration in the introduction, we have added some elaboration in conclusions (line 297 -300).

The details are as follows:

“These investigations, based on microscopic understanding, elucidates the relation of energy current and quantum coherence, which might potentially be a good reference in context of quantum thermodynamics of non-Markovian open quantum systems, as well as in study of environment-induced quantum coherence.”

Round 2

Reviewer 1 Report

I have gone through the revised manuscript entropy-1929154. The authors adequately responded to my comments. I appreciate it that they included more information about the energy current. Also, some technical errors related to the figures and references have been eliminated. Finally, I have checked the other referees' reports and I see that the manuscript has improved due to those suggestions. Overall, I think that the study devoted to energy current under non-Markovian environments is a valuable contribution to the theory of open quantum systems. The manuscript deserves to be accepted in the present form.